# Application of AI Identification Method and Technology to Boron Isotope Geochemical Process and Provenance Tracing of Water Pollution in River Basins

Gang Hou, Hui Yan *  and Zhengzheng Yu

College of Urban and Environmental Sciences, Xuchang University, Xuchang 461000, China
* Correspondence: 12009002@xcu.edu.cn

**Abstract:** River water is the most important water source that people can use. Since the 20th century, human influence on river courses has become increasingly serious. The quantitative analysis of water quality is even more difficult. According to the characteristics of Fenhe water chemistry, pollution time and pollution control factors, the contribution rate of people in the polluted water body is not clear. Therefore, this paper aims to use AI identification methods and technologies to study water pollution and provenance tracing. The combination of major elements, trace elements and stable isotopes was used to study the chemical characteristics, water quality status, and sources of pollution of the Fenhe water in the Fenhe area. Because the water contains a large number of pollution sources, it is difficult to find the source using traditional methods. Using correlation analysis, principal component analysis, multi-factor regression analysis, trend analysis and other methods, the macroelements and trace elements in the water body of the Fenhe River were analyzed. The boron sources in the Fenhe river were qualitatively and quantitatively analyzed using mass spectrometry equilibrium equation. Using the boron isotope value of the river, it showed a spatial variation of upstream (+5.1‰) < middlestream (+8.6‰) < downstream (+9.5‰) in dry season, and showed a spatial variation of upstream (+6.1‰) < downstream (+7.2‰) < middlestream (+9.0‰) in the wet season. The contribution of silicate to B is calculated by subtracting the contribution of other resources from the comprehensive contribution rate. It is found that the contribution of silicate is about 38.8%, 22% in dry season and 49.2%, 17% in wet season. The research results have provided a reliable scientific basis for the protection of water resources and pollution control in the Fenhe River Basin. Therefore, the above research confirms the role of AI identification method in the process of boron isotope geochemistry and provenance tracing of water pollution in river basins.

**Keywords:** water pollution; boron isotope; provenance tracing; AI identification method

## 1. Introduction

Water resources have received great attention in China's economic and social development. However, China's water ownership rate is far below the international average. The dissemination of water assets is additionally extremely lopsided. As the industry keeps developing, the different effects individuals have on the waterway framework are additionally expanding. The pollution of river water not only causes serious harm to the ecology, but also endangers the physical and mental health of the residents. Toxic substances in river water can enter the body through food and other means. Toxic substances pollute water bodies. When various toxic substances enter the water body, they will kill the aquatic organisms at a high concentration. At low concentrations, it can be enriched in the organism, and gradually concentrated through the food chain, finally affecting the human body. Once accumulated to a certain critical point, certain symptoms are triggered. After space research, it was found that, it has a large number of pollution sources and produces a series of physical, chemical and biological effects after entering the water. It is difficult to trace the source effectively using traditional methods with other isotopes. In

recent years, with the rapid development of stable isotope geochemistry, stable isotopes have been widely used as tracers in the aquatic environment. Due to the large relative mass difference of B isotopes, the boron isotope composition is extensive. Boron and boron elements vary in different rock formations, which can serve as a good indicator. In this paper, the chemometrics method in the artificial intelligence method is used for qualitative and quantitative analysis. This paper has certain reference significance for the later study of boron isotope or provenance tracing.

Accurate water source identification is the best choice for water quality management strategies. In Hu M's research, three years were used to conduct a comprehensive analysis of various stable isotopes in rivers, groundwater, rainfall, etc., to reveal N dynamics and the origin of China's Yongan Basin. The main pollutants of non-point source nitrogen were pollutants in soil. Their forms and contents had obvious spatial and temporal variation [1]. Xin-Qiang discussed the main pollution sources and geographic locations of pollutants in urban groundwater MIN in China. Due to the diversification of nitrogen sources and the complex pollution mechanism of MIN, the scope of application of a single method was limited. A more comprehensive approach was adopted. In particular, the stable isotope tracing technique used in combination with other methods had become the main research direction [2]. The purpose of the Guinoiseau D test was to determine the content and isotopic properties of boron in the Seine over 18 years. He explored whether boron isotopes could track human input over time. In the artificial Seine Valley, the boron content was below the drinkable limit due to mass production by humans, which proved that the radioactive element of boron could track artificial pollution in the city [3]. Zhang Y described the use of backward probability density function (PDF) to analyze the source and release time of superdiffuse attenuated pollution. Parametric analysis was utilized to revisit the well-known MADE-2 tracer test results. The results showed that this method could predict the reverse direction PDF [4]. Efthimiou G C evaluated the state-of-the-art inverse source parameter estimation method and validated it in two model tests, which were issued under real scenarios, namely Michelstadt and CUTE. This approach appeared to be reliable and can accurately predict source points and emission rates in two wind tunnel tests [5]. In sudden water pollution accidents, the transport and diffusion of pollutants are restricted by factors such as terrain and complex water environment. This makes it uncertain in time and space, which makes it difficult to mathematically simulate changes in pollutants in complex environments.

Without relying on factors such as conductivity, porosity, and hydraulic gradients, Cremeans M M used the SBPVP method to determine the subsurface flow velocity at the boundary between the surface and groundwater bodies. The SBPVP data were compared with water level and temperature gradient data collected from a similar scale. Studies had shown that in local high-velocity areas, the discharge of pollutants into the river was not necessarily compatible with the maximum pollution [6]. In order to study the effect of quantitative calculation of pollutant source load on the degree of pollution damage, Zhao P proposed two typical methods of classification index and quantitative index. He linked it to the same vulnerability and groundwater value, and compared the establishment of various risk assessment systems [7]. The purpose of Jeon J was to provide an effective and cost-saving detection method for the establishment of an automatic detection device without point source pollution. The study found that regardless of the rainfall, long-term observations could be made in the forest belt and the downstream areas [8]. Taking Sanbanxi Reservoir as an example, He W conducted an empirical study using the Flow-3D three-dimensional hydrothermal model. He also carried out qualitative and quantitative analysis of dead weight ton (DWT). Among the water layers, the largest active layer was located 7.5 m above the water inlet [9]. Nitrogen sources were determined by Gulcicek O using isotopic analysis combined with groundwater models. The source of nitrogen gas was determined through stable isotope determination of 15 N. A numerical simulation of the sedimentary aquifer near the estuary was carried out using the 3D model of Visual MODFLOW. After the construction of MT3DMS (MT3DMS), the nitrate nitrogen in soil

was studied using particle tracking technology [10]. Deterministic algorithms can be used to obtain optimal results under limited data conditions. However, it also leads to the distortion of parameters, which affects the reliability of the retrospective modeling of the product. Measurement uncertainty is a quantitative representation of the quality of measurement results. The availability of measurement results largely depends on the size of its uncertainty. Therefore, the expression of measurement results must include both the measured value and the measurement uncertainty associated with the measured value to be complete and meaningful.

The results showed that the boron isotope content of rivers in arid area was +0.6~14.9%, +7.61%, and the difference rate was 47.8%. During the wet season, the amount of boron in the river was between +0.6 and 13.1, with an average of −7.94 and 42.7%. The results were shown in the dry water period; the boron isotope content of the river showed the upstream (+5.1‰) < middle (+8.6‰) < downstream (+9.5‰) in the dry water period and the upstream (+6.1‰) < middle (+9.0‰) in the wet season.

In this paper, major elements, trace elements, stable isotopes and other elements were used to analyze the pollution status of the Fenhe Fenhe River Basin. Through the analysis of the pollution status of the Fenhe River, the water pollution of the Fenhe River Basin could be effectively controlled, thereby effectively controlling the water quality of the river and promoting the healthy and stable economic development of the Fenhe River Basin.

## 2. Geochemical Process of Boron Isotope in Water Pollution and Provenance Tracing Method

### 2.1. Artificial Intelligence (AI) Technology

AI is a part of computer technology that produces a new type of intelligent AI, which can understand the nature of intelligence and respond in the same way as human intelligence. With the passage of time, its principles and technologies have been gradually improved, and its uses have continued to expand. In the future, the technology generated by AI could become the "container" of human intelligence. It certainly promotes the development of science and technology [11]. Artificial intelligence is an extremely challenging subject. In general, an important purpose of artificial intelligence is to enable computers to perform certain complex tasks.

Regarding the concept of intelligence, there is no doubt about the connection between a computer and intelligence. Among them, the most prominent performance is the processing of AI and intelligent information [12]. Between neuroscience and intelligence, it can be divided into two levels, namely, the structural level and the operating mechanism level of the system, among which the behavior and cognition of intelligence are the main ones.

Computerized reasoning innovation is an exceptionally wide subject. Because of the turn of events and improvement of brain science, physiology, arithmetic, reasoning and different fields, the extent of its examination and application has been constantly extended. Groundbreaking thoughts, hypotheses and advancements are continually arising. The scope of AI applications is shown in Figure 1.

As shown in Figure 1, the current development of artificial intelligence has been widely used in life, including industry, service, unmanned driving, medicine, finance, education, judiciary, military, and so forth. However, it is still in the weak AI stage. It is called "weak" because it does not have the ability to think, reason and solve problems, which is not a kind of wisdom [13]. In contrast, a powerful artificial intelligence can perceive, think, and act independently as long as it works in conjunction with the appropriate programming language.

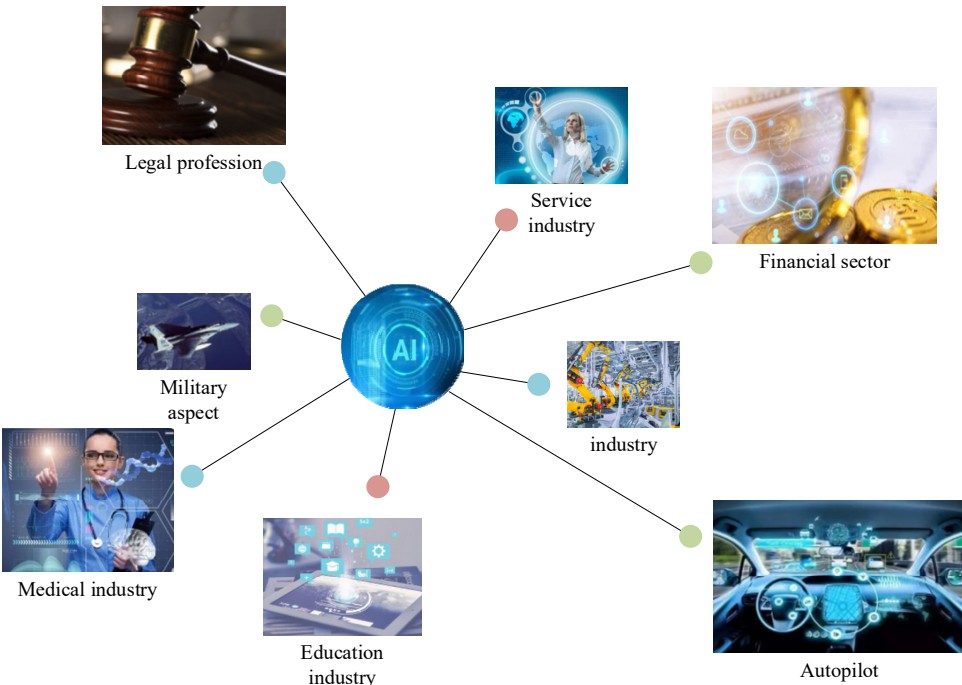

**Figure 1.** Application of AI technology.

*2.2. Convolutional Neural Networks*

A typical convolutional neural network consists of a series of processes where locally weighted sums are passed to nonlinear functions. This paper uses convolutional neural network technology to track boron isotopes. The basic structure of the convolutional neural network is shown in Figure 2.

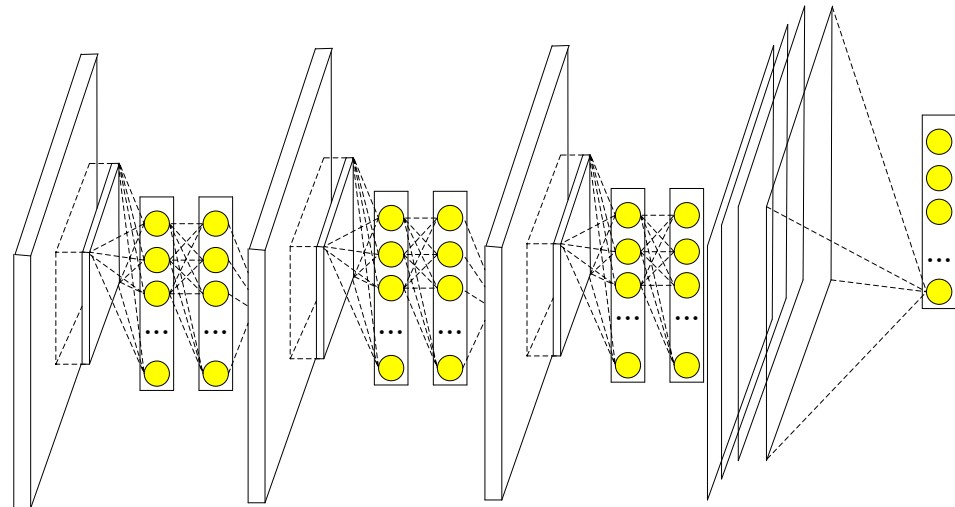

**Figure 2.** Basic structure of convolutional neural network.

As shown in Figure 2, the function of convolution is essentially to extract the features of the region. The pooling level is to comprehensively process the features with semantic similarity. On the feature map, the pooling level is generally used to find the largest local block. Adjacent central nodes use one or more rows for data processing [14]. The sampling diagram is shown in Figure 3.

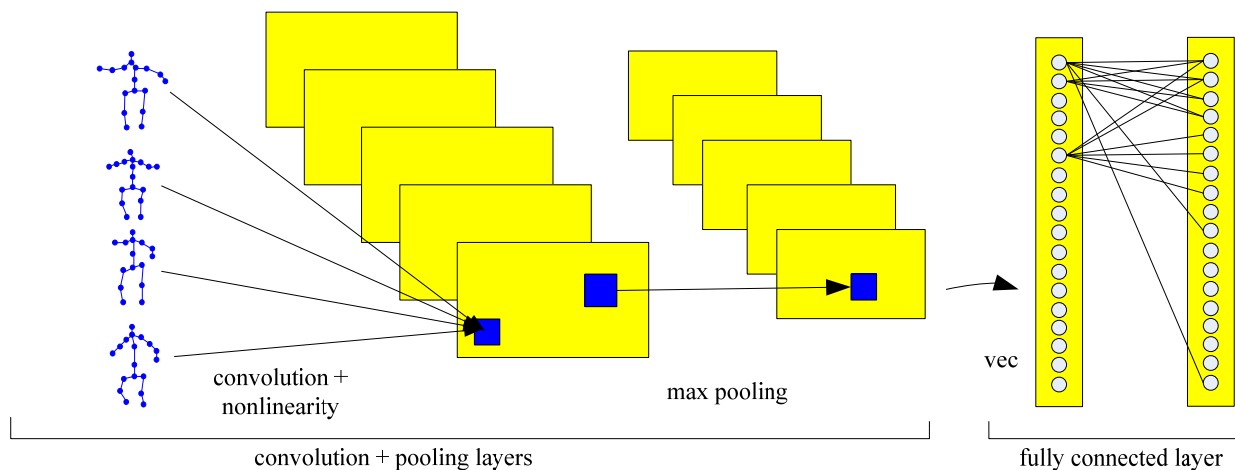

**Figure 3.** Sampling schematic.

As shown in Figure 3, the sub-sampling layer performs maximum sampling, which selects the maximum value in each sub-region as the sub-sampling result. The nonlinear sigmoid function and function gradient are shown in Figure 4.

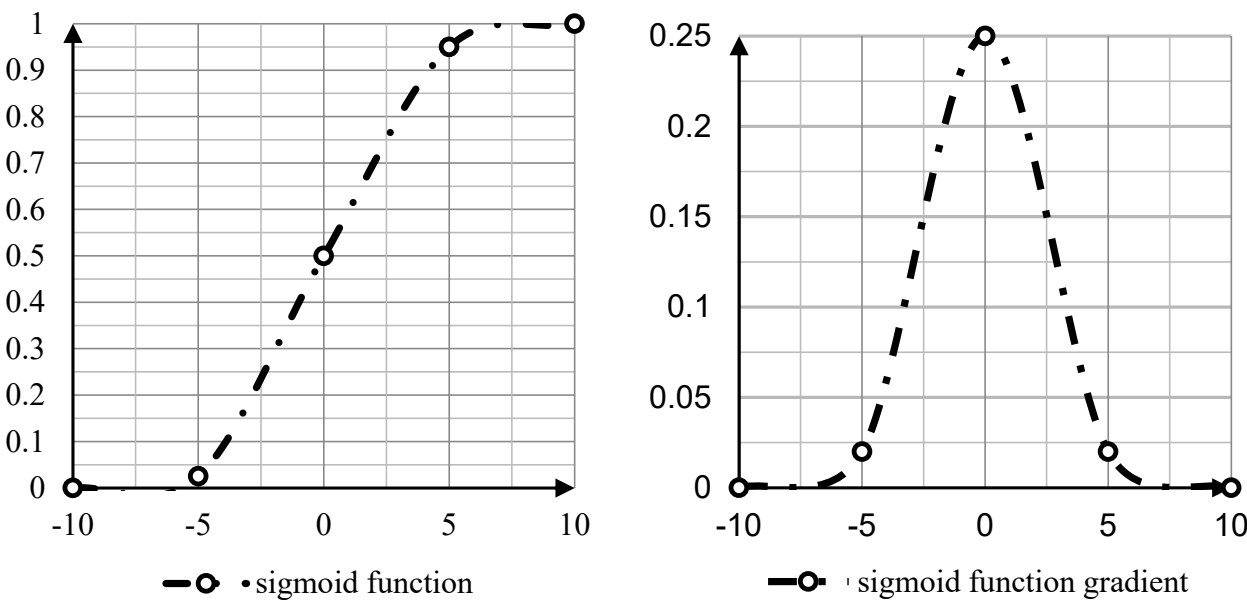

**Figure 4.** Sigmoid function and gradient.

As shown in Figure 4, the function value of sigmoid is between 0 and 1 in the open interval. The closer the function value of the independent variable is to 0, the faster the function value changes. The larger the absolute value of the independent variable, the slower the function value changes. The function and gradient of ReLu are shown in Figure 5.

As shown in Figure 5, in the classic structure, the activation function using the neural network is called the rectified function. In order to alleviate the "dead zone" phenomenon, the part of $x < 0$ in the ReLu function is adjusted to $\psi(x) = \delta * x$. Among them, a is a small positive number on the order of 0.01 or 0.001 [15].

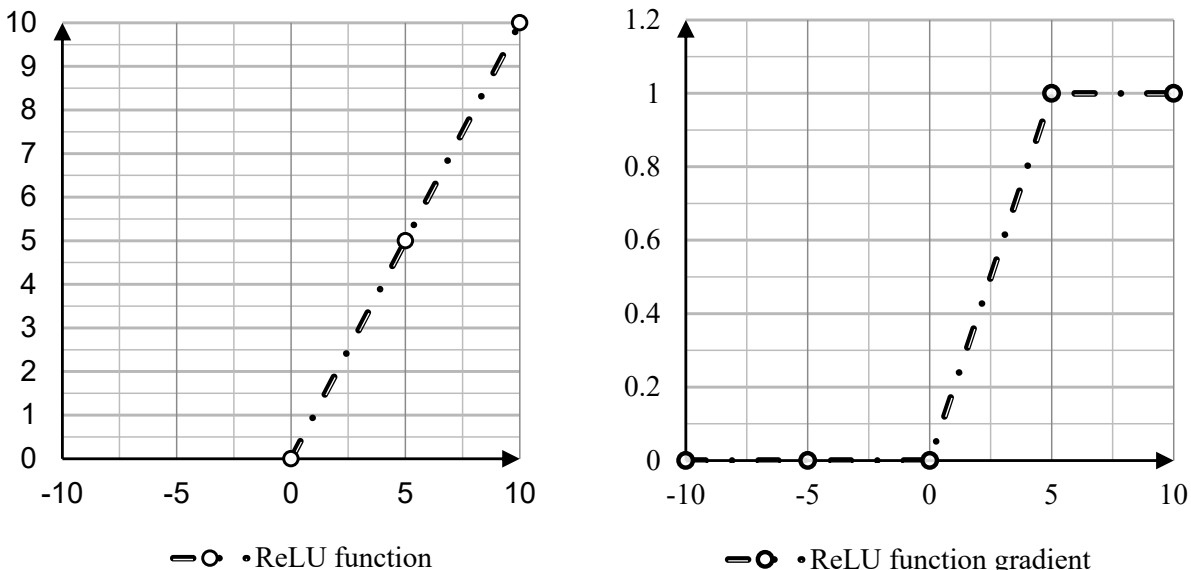

**Figure 5.** ReLu function and gradient.

### 2.3. Determination Methods and Geochemical Progress of Boron Isotopes

The atomic number of boron is 5. Its properties are the stability of 11B and 10B: 10B = 19.9%, 11B = 80.1%. Because of the large mass difference of boron, boron isotopes are contained in many chemical reactions and geological processes in nature [16]. Boron isotopes have a wide range of uses, and their main function is that the boron content of each ore body in nature varies greatly. In different geochemical libraries, not only are there different boron isotopic compositions, but the boron content is also very different. These two methods can be combined for traceability studies. There exist the chemical exchange distillation method, carbon-alkali method, soda ash solution (sodium borosilicate), etc. After the boron in the sample is purified and separated by the commonly used boron specific resin method, there is still Cl—an ion introduced in the solution in the form of HCl during leaching. Generally, HCl is removed through evaporation. Since the boron content in the air is low, the background value of the determination process is easy to carry out. Boron is a very common element that exists widely on Earth, such as in salt lakes, oceans, and sedimentary rocks, which are the main sources of boron ore [17]. The boron isotopic composition $\sigma^{11}(B)$ (%) can be represented as in the following equation:

$$\delta^{11}\text{B}(\%) = [(^{11}\text{B}/^{10}\text{B})_{\text{sample}}/(^{11}\text{B}/^{10}\text{B})_{\text{standard}} - 1] \times 1000 \tag{1}$$

The isotopic composition of natural boron varies widely around 90%. The content of 81 B is about $-36$ thousandths, mainly distributed in non-ocean evaporative salts and some calcium carbides. In the Dead Sea and the Australian Salt Lake, 81 B is a feature of the late magmatic period. Area B is brought to the surface by factors such as groundwater thermal system and surface runoff [18]. Through the infiltration test of magma, it is found that boron can be flushed out of the rock mass when it is lower than the melting temperature of the rock mass. Even at 20~22 degrees Celsius, 50% of magma B can be leached out. Under the B-type water-rock interaction, the isotopic content of boron is essentially the same [19].

### 2.4. Surface River Water Quality Model

With the continuous implementation of the water pollution prevention and control plan, China's ecological environment has gradually improved in recent years. From 2014 to 2018, the water quality of China's controlled surface water and the water quality of the ten major river basins are shown in Figure 6.

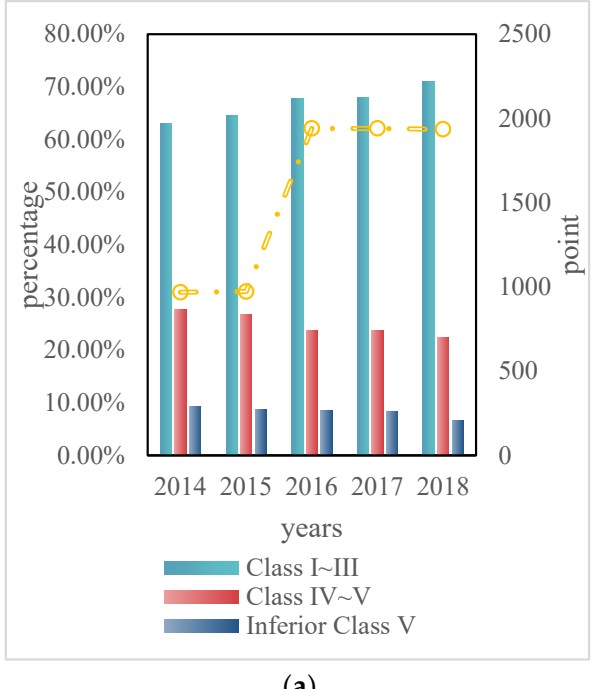 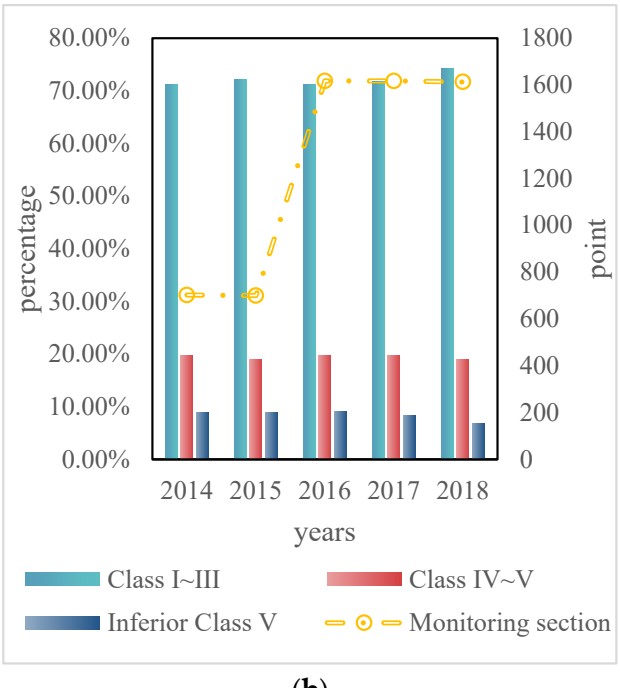

(a)　　　　　　　　　　　　　　　　　(b)

**Figure 6.** The water quality of China's controlled surface water and the water quality of the ten major river basins from 2014 to 2018. (**a**) The water quality of China's controlled surface water. (**b**) The water quality of China's controlled sections in the ten major river basins.

As shown in Figure 6a, the water quality of grades I to III of China's controlled surface water quality has gradually increased. As shown in Figure 6b, the water quality of the V-level control section is diminishing. Overall, China's river water pollution has generally declined. Compared with the previous decade, the severely polluted water quality has been greatly improved [20]. However, there is still a long way to go to achieve the overall improvement of the national water environment quality and the basic restoration of water ecological functions by 2030.

After a water pollution event, the movement behavior of pollutants in the surface reaches conforms to the convection–diffusion equation. The inverse problem that the water pollutant term in the surface reaches is based on the water dynamics–water quality coupling system. The incoming water from the lake basin is the main dynamic factor controlling the change of sediment particle size composition. According to the principle of sediment mechanical sorting, the size of sediment particles gradually becomes smaller from the lake edge to the center of the lake, and they are distributed in concentric circles. At different locations in the lake, the size of sediment particles can change with the change in transport force.

Its governing equations are shown in Equations (2)–(4):

$$\nabla f = 0 \tag{2}$$

$$\frac{\partial f}{\partial t} + f \nabla f = -\frac{\nabla \alpha}{\alpha} + g \nabla^2 f \tag{3}$$

$$\frac{\partial A}{\partial t} = -f \nabla A + \nabla (D \nabla A) - kA + R(m, n, z, t) \tag{4}$$

To further measure the pollution in water, it is necessary to clarify its concentration and initial concentration distribution at the boundary of the water body. It is supposed that the sea area under investigation is Q. T is the boundary of the water body, including two regions T1 and T2 [21]. The initial distribution of pollutants in the water area Q is $A_0(m, n, z)$. Its mathematical expression is shown in Equation (5):

$$A(m, n, z, t)|_{t=0} = A_0(m, n, z) \tag{5}$$

The concentration field distribution function $f(m, n, z, t)$ is given at each point of the boundary Tn, as shown in Equation (6):

$$A(m, n, z, t)|_{\Gamma_1} = f(m, n, z, t) \tag{6}$$

The concentration flux function $g(m, n, z, t)$ is given at each point of the boundary T2, as shown in Equation (7):

$$\sum_x \left( f_x A - D_x \frac{\partial A}{\partial x} \right) \cos(n, x) \bigg|_{\Gamma_2} = g(m, n, z, t) \tag{7}$$

In light of the above suppositions: in the stream express, the parts of $f$ and $D$ in the $m$, $n$, and $z$ bearings are unaltered. The emission of pollutants is decomposed into a first-order dynamic mass loss, and combined with biochemical reactions, which is expressed by the overall decomposition factor k of the pollution [22]. Based on the one-dimensional hydrodynamics-water quality model, numerical simulations are carried out. Therefore, Equation (4) is simplified into Equation (8):

$$\frac{\partial A}{\partial t} = -f_i \frac{\partial A}{\partial i} + D_i \frac{\partial^2 A}{\partial i^2} - kA \tag{8}$$

For the one-dimensional hydrodynamic-water quality model of the apogee, it is assumed that the pollution source is at a definite location and a definite location as the starting point of a spatial coordinate. The time of the initial release of a pollutant is determined as the origin of a time [23]. The pollutant mass changes with the action of convection, diffusion, etc., in the measured water body [0, L] and the time range [0, T]. Among them, the emission history function of the pollutant is s(t). The concentration at infinity x = L is 0 [24]. Its initial and boundary conditions are divided into Equations (9)–(11).

$$A(i, 0) = 0 \tag{9}$$

$$A(o, t) = s(t) \tag{10}$$

$$A(L, t) = 0 \tag{11}$$

The exact solution of this problem is shown in Equation (12):

$$A(i, t) = \int_0^t s(\omega) \frac{i}{2\sqrt{\pi D_i(t - \omega)^3}} \exp\left\{ -\frac{[i - f_i(t - \omega)]^2}{4D_i(t - \omega)} - k(t - \omega) \right\} d\omega \tag{12}$$

Among them, $\omega$ is the continuous discharge time of pollutants. $D_i$ is the longitudinal diffusion coefficient of the river. $f_i$ is the longitudinal velocity of the river.

### 2.5. Pollution Source Method

In the source solution analysis model, the polluted river channel is mainly used as the model. The models mainly include the chemical mass balance model and the multivariate statistical model [25]. Using the PCA-APCS-MLR method, the primary sources of trace elements in the Fenhe River are analyzed. Using a chemical equilibrium model, borides in

the river are studied. The MixSIAR model is used to study nitrate nitrogen in rivers. The PCA-APCS-MLR is operated with sPSS23.0.

In this paper, the causes of dissolved boron are divided into three types: rainwater input, anthropogenic input and rock weathering. Rock weathering can be divided into carbonatite weathering, evaporite dissolution and silicate weathering [26]. For wind erosion of silicate rocks, the total contribution is used. Together with other end members, the calculation equation for rainwater input is:

$$C^* = (C/CI)_{rain} \times CI_{ref} \tag{13}$$

$$CI_{ref} = F \times CI_{ave} \tag{14}$$

$$F = P/(P - E) \tag{15}$$

In Equation (15), P is the annual average precipitation (mm). *E* is the annual average evaporation (mm). The average B/CI molar ratio of rainwater samples from the Fenhe River Basin is 0.021. Among them, the equations entered by humans are shown in Equations (16) and (17).

$$[B]_{anth} = [Na]_{anth} \times [B/Na]_{anth} \tag{16}$$

$$[Na]_{anth} = No_3^-{}_{river} \times [Na/No_3^-]_{anth} \tag{17}$$

In the equations, $[B]_{anth}$, $[Na]_{anth}$, and $[B/Na]_{anth}$ represent the artificially input B in the river water, the artificially input Na in the river water, and the molar ratio of the artificially polluted endmembers B/Na, respectively. Among them, the carbonatite weathering is shown in Equation (18) [27].

$$[B]_{carbonates} = [Ca]_{river} \times [B/Ca]_{carbonates} \tag{18}$$

In Equation (18), $[B]_{carbonates}$, $[Ca]_{river}$, and $[B/Ca]_{carbonates}$ represent $B$ input from carbonatite weathering in river water, the concentration of $Ca^{2+}$ in river water, and the molar ratio of carbonatite endmember B/Ca, respectively [28]. For example, the evaporite dissolution equations are shown in Equations (19) and (20).

$$[B]_{halite} = [CI]_{river} \times [B/CI]_{halite} \tag{19}$$

$$[B]_{gypsum} = [SO_4]_{river} \times [B/SO_4]_{gypsum} \tag{20}$$

## 3. Experiment of Boron Isotope Geochemical Process and Provenance Tracing

### 3.1. Overall Design of Water Pollution Traceability Management System

In this paper, the artificial intelligence algorithm and boron isotope are used to study the traceability of polluted elements. According to the structure in the software development process, the overall design of the river sudden water pollution traceability management system is carried out. The system consists of four layers: user layer, business layer, support layer and data layer. Its specific structural frame diagram is shown in Figure 7.

As shown in Figure 7, the user level responds to the user's request and displays it as a human–machine interface. The user interface of the system consists of eight modules: user login interface, system main interface, data acquisition interface, data filtering interface, pollution tracing interface under artificial intelligence, result display interface, result output recording interface and system management interface. The main modules of the system include: data collection, data filtering, pollution traceability, result visualization, result output record, and system management. In addition, for different data and results, the service level can also perform functions such as query, add, delete, modify, clear and reset.

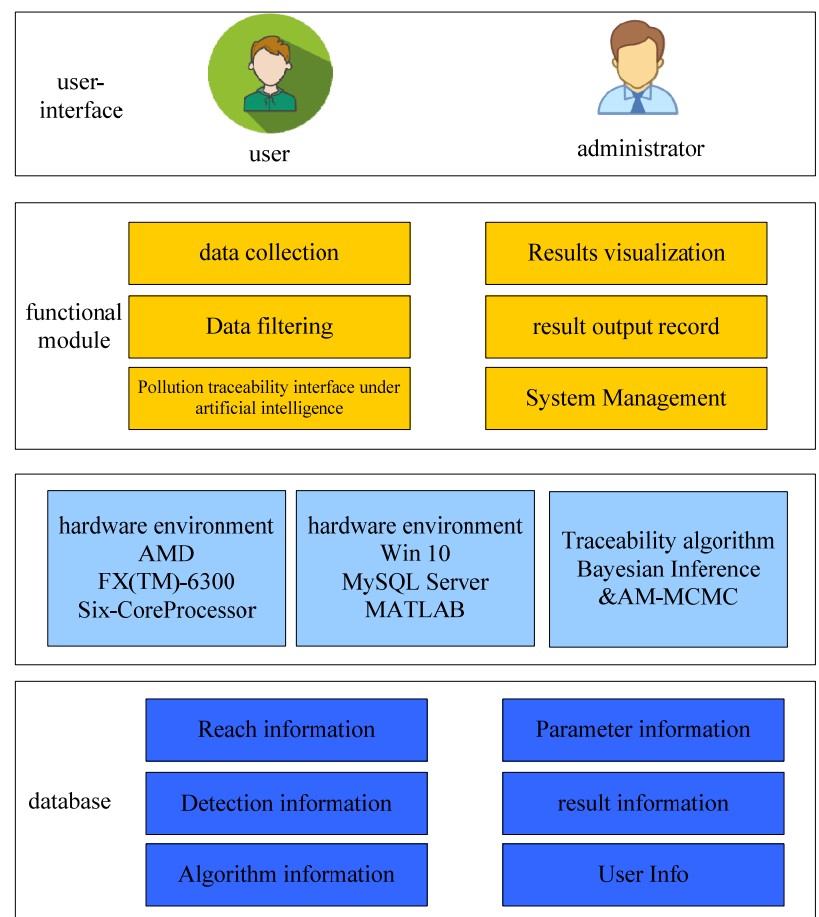

**Figure 7.** Structural framework of water pollution traceability management system.

*3.2. Quantitative Analysis of the Sources of Pollutants in the Fenhe River Basin*

Depending on the intensity and duration of rainfall, the study should last at least one year to study the trends or patterns in the physicochemical properties of traceable elements, influencing factors of pollutants, etc. Boron has important uses in industry, fertilizer, detergent, metal welding, insulation, aerospace and other fields. In addition, the solubility of B in river water is also affected by some natural factors, such as evaporite, carbonatite, silicate rock, etc. Boron isotope has the ability to trace the process of water–particle interaction; therefore, the physical and chemical weathering in the process of water and soil loss in the basin can be studied using the theory of boron isotope geochemistry. It can enrich the boron isotope geochemical theory of surface water–rock interaction process, and provide scientific evidence for understanding the erosion process. The standard boron isotope abundance ratio is $4.04362 \pm 0.0013710$. B has high solubility and relatively conservative chemical properties in rivers, which is easy to accumulate in rivers. In fact, Class B pollutants exist on a global scale. For example, the levels of B in Himalayan rivers, Kao-ping river, etc., are much higher than those in major global rivers (25 ug/L). The pollution of river B poses a huge threat to the river's ecology and biological life.

First, a qualitative study was conducted on the origin of dissolved B in the Fenhe River. In the birthplace of the Fen River, the boron isotope content of the Honghe River is +8.3%, +9.0% in the wet season, and +6.4% in the Daming River section, +8.8% in the wet season. The correlation between boron isotopes and CI/B in the water of the Fenhe River Basin is shown in Figure 8.

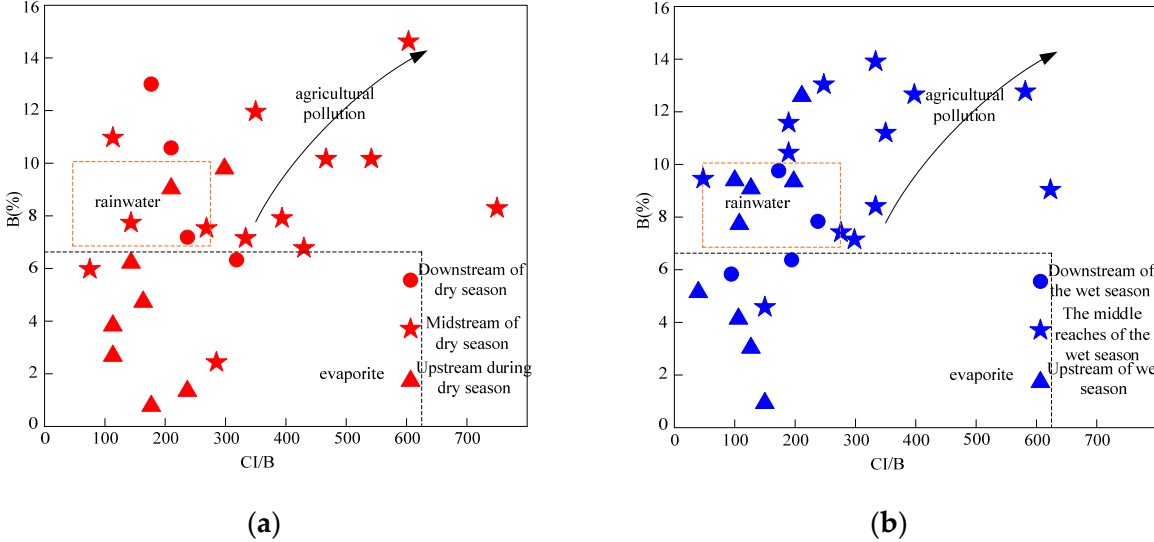

**Figure 8.** Correlation diagram of boron isotopes and CI/B in the waters of the Fenhe River Basin, (**a**) CI/B value in the upper, middle and lower reaches of the dry season, (**b**) CI/B value in the upper, middle and lower reaches of the wet season.

As shown in Figure 8a, the qualities of precipitation close to the water wellspring of the Fenhe Stream demonstrate that its boron content is firmly connected with precipitation. Upstream, near the end members of the gasification rock, the boron content is very low. Such rivers can be classified as mild or mild B. There is also a large amount of boron used in the agricultural field. It can be seen from Figure 8b that the agricultural sewage is mainly fertilizer and livestock excrement, and its CI/B and NO/B values are high. The average level of agricultural pollutants is higher than that of industrial pollutants. The content of boron isotopes in different types of farmland wastewater is in a wide range. For example, boron isotope levels in farmland wastewater in the United States range from −2.0% to +0.7%. France is between −8.0% and +7.0 one thousandth. Figure 9 shows the correlation between boron isotopes and Na/B molar ratio in the water of the Fenhe River Basin.

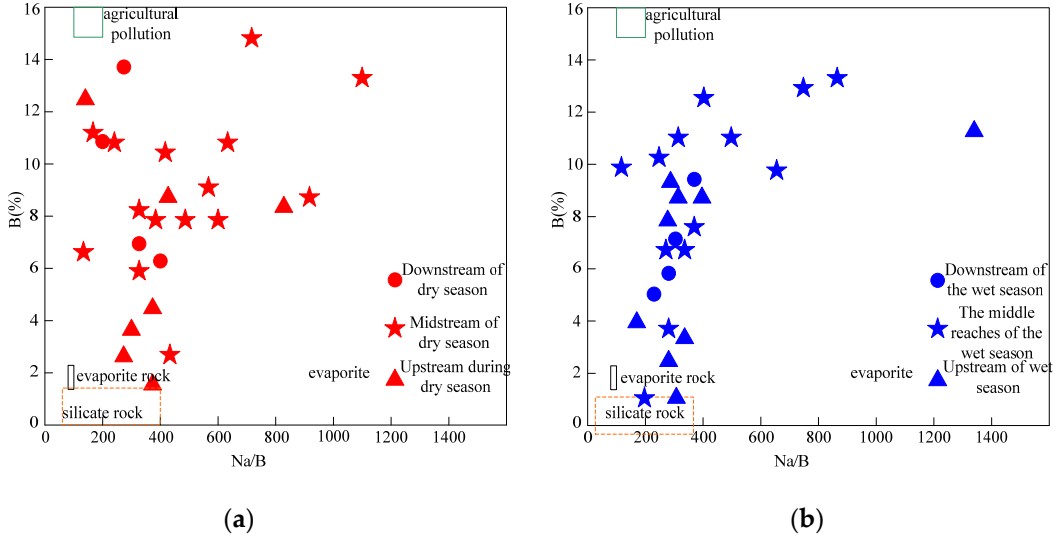

**Figure 9.** Correlation diagram between boron isotopes and Na/B molar ratio in the water body of the Fenhe River Basin, (**a**) Na/B value in the upper, middle and lower reaches of the dry season, (**b**) Na/B value in the upper, middle and lower reaches of the wet season.

As shown in Figure 9a, the results show that the water quality of the central reaches is similar to the end elements of agricultural pollution, indicating that the agricultural environmental pollution in this region has a certain contribution to the water quality of the watershed. Some rivers in the lower reaches of the Fen River, as shown in Figure 9b, have a boron isotope content of +11.35% during the wet season. Among them, the contents of CI and SO42 are high, indicating that they are closely related to the introduction of agriculture.

The production of boron in China is dominated by borate ore. There is no significant isotopic separation in production. Its boron isotope content is between −19.5 and 11.1%. The results show that in urban domestic wastewater and domestic wastewater, B element mainly comes from boron ore. The boron isotope content should be at the same level. Urban wastewater is higher than general industrial wastewater, which is caused by differences in pollution sources. Some rivers in the middle reaches of the Fen River (such as the Xiangyu River) have higher concentrations of $NO_3$ and B. The boron isotope value is low and may be affected by anthropogenic pollution. Figure 10 shows the correlation between boron isotopes and the $NO_3/B$ molar ratio in the water of the Fenhe River Basin.

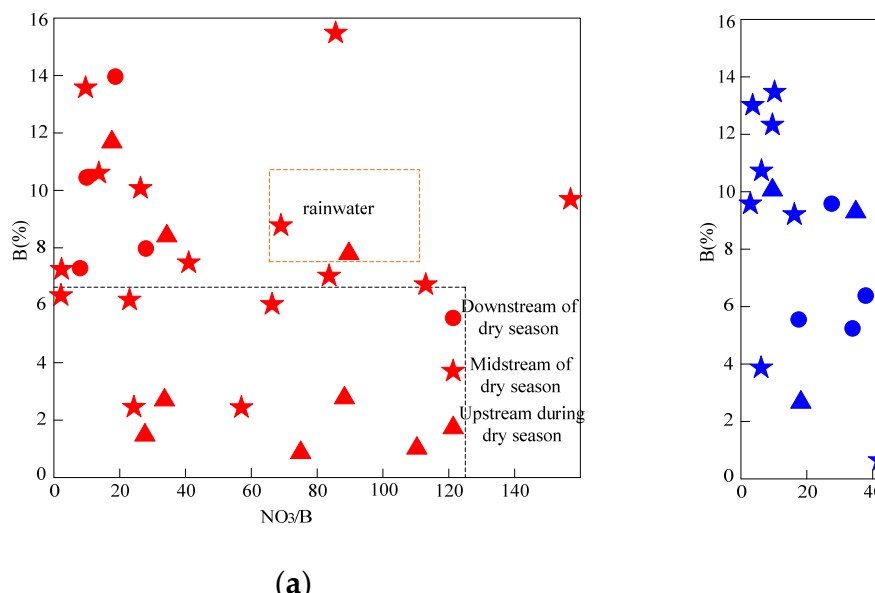

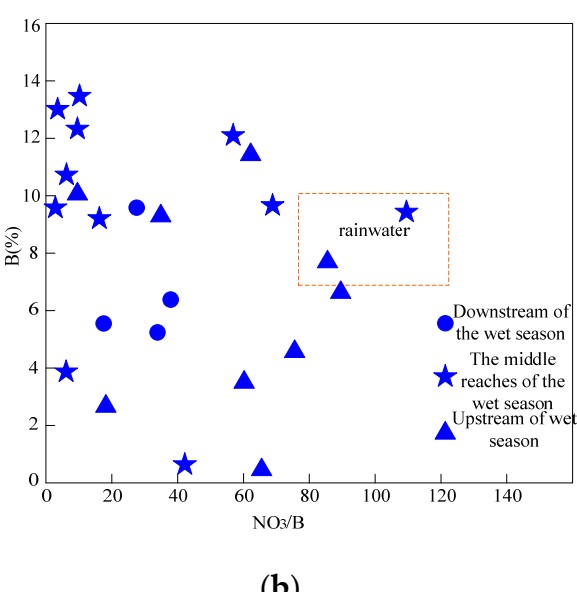

(**a**)                                  (**b**)

**Figure 10.** Correlation diagram between boron isotopes and $NO_3/B$ molar ratio in the water body of the Fenhe River Basin, (**a**) $NO_3/B$ value in the upper, middle and lower reaches of the dry season, (**b**) $NO_3/B$ value of upstream, middle and downstream during wet season.

As shown in Figure 10a, the boron isotope content of the two major rivers in the Fenhe River, Huihe and Jianhe, decreases during the wet season. However, the value of $NO_3/B$ also increases to a certain extent. This indicates that the content of boron isotopes in the main water body of the Fenhe River has increased. Figure 10b shows that the $NO_3/B$ and CI levels in some reaches of the Fenhe River are relatively low. The abnormality of B-isotope is related to the mixing of groundwater. Therefore, there are three main types of borides in soils in the Fenhe River Basin: rainwater input, anthropogenic input, and soil erosion.

The calculation results of the contribution of each pollution source (rainwater input, human input and rock weathering) of dissolved boron in the river water of the Fenhe River Basin are as follows.

(1)   Rainwater input

Although the concentration of B in rainfall is relatively small, it also has a certain effect on some waters with less B. In this experiment, the collected rainfall samples have concentrations of B between 0.82 and 1.99 ug/L, which is approximately 1.59 ug/L. The

boron content is between +6.7–9.9% and +7.9%. During the dry season, the contribution of rainfall to B is between 0.7–33.0% and 7.8%. The contribution of B in the rainy season ranges from 0.7% to 20.6% and 6.7% of the wet season. The spatial and temporal distribution characteristics of rainfall input are as follows: the average level of rainfall input in the dry season of the Fenhe River Basin is 12.9%. The precipitation input in the middle and lower reaches is 4.9%, and the downstream is 2.8%. During the wet season, the average values of the upper, middle and upper reaches are 11.8%, 3.2% and 3.2%, respectively. This indicates that rainwater input would affect the concentration of B.

(2) Human input

The study found that in the arid area of the Fenhe River, the contribution of artificial input to B is 16.4–14.8% (1 SD), 15.0 ± 9.8% (1 SD). The artificial input in the Fenhe area is close to that in the Yangtze River area. The results show that the artificial input has a great effect on the dissolution of B in the Fenhe channel.

(3) Rock weathering

Assuming that all the calcium in the river water comes from carbonatite, the estimation of the contribution of carbonatite to B shows that the contribution of carbonatite to B is 2.6 ± 2.2% (1 SD) in the wet season and 2.2 ± 1.8% (1 SD) in the arid region. Overall, the contribution of carbonate content in rivers B such as the Yangtze (<3%) and Himalayan River (~5%) is almost nil. The contribution of different pollution sources to dissolved boron in the water of the Fenhe River Basin is shown in Figure 11.

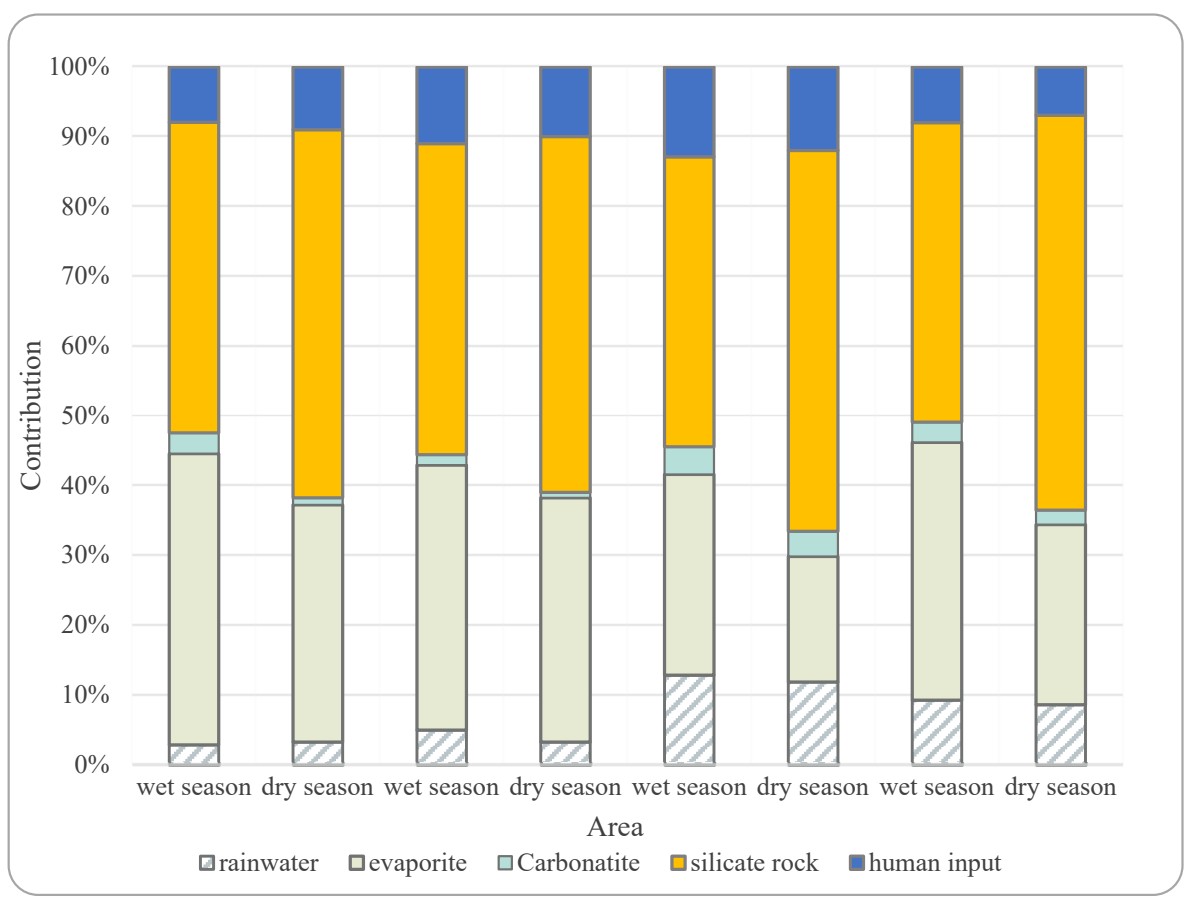

**Figure 11.** Contribution of different pollution sources to dissolved boron in the water of the Fenhe River Basin.

As shown in Figure 11, through qualitative research on borides, it is found that some samples are close to the endmembers of vaporized rocks. Therefore, the dissolution of

evaporites plays an important role in river water B. In the dry season, the contribution of dissolved soil salts to B is 32.9 ± 13.5% (1 SD). The upstream, midstream and downstream contribute 24.0%, 32.0%, and 40.0%. During the wet season, the contribution of salinity to B is 23.7 ± 15.1% (1 SD). The upstream, midstream and downstream contribute 16%, 31%, and 30%, respectively. The study found that in the rich water period, the contribution of gypsum dissolution to B is 3.1–4.3%. Shahe accounts for 0.1%. The interactive river contributes the most (23.4%). In the dry season, the contribution to B reaches 4.3 ± 5.9% (1 SD). In arid regions, the contribution to the dissolution of gypsum is essentially the same as that in the wet season. Shahe accounts for 0.1%. Sediment contributes the largest, accounting for 31.7%. The general view is that sediment and sediment dissolution can reflect the effect of total evaporation on B. The results show that the contribution of evaporite ablation to B is 37.3% and 27.1% in arid and wet areas, respectively. Silicate minerals do have a certain influence on the content of B. In this paper, the contribution of silicate to River B is estimated by deducting the contribution of other resources from the comprehensive contribution rate. The results show that silicate contributes 38.8% ± 22% (1 SD) to the soil during dry periods and 49.2% ± 17% (1 SD) during wet periods.

### 3.3. Comparison of Sampling Covariance Update Methods

Using standard, fixed step size (h = 100), memory matrix (H = 5 h), mixed (h = 100, H = 5 h), and the method of operation parameters, on the basis of the unrelated homoscedasticity assumption, three different contaminants are traced. The influence of the covariance matrix update method on the instantaneous point source is shown in Table 1.

**Table 1.** Effect of covariance matrix update method on instantaneous point sources.

| Index | Unit | Standard | Fixed Step | Memory Matrix | Mix |
|---|---|---|---|---|---|
| Proposed sample total acceptance rate tar | % | 50.49 | 31.71 | 60.75 | 60.48 |
| Iteration time-consuming eti | s | 311.53 | 21.35 | 20.64 | 18.49 |
| Execute time-consuming ete | s | 696.76 | 356.27 | 376.84 | 377.85 |

It can be seen from Table 1 that in the real-time traceability simulation of the point source, the iteration time of fixed step size, memory matrix and chaotic optimization algorithm is reduced by 93.15% compared with the traditional method, and the memory matrix is reduced by 94.06%. The results showed that the completion time is reduced by 47.21%, 43.68%, and 45.98%, respectively. The other two methods are in the higher interval, except for the fact that the method with fixed-step update reduces the overall acceptance proportion of sampling. The effect of the covariance matrix update method on continuous point sources is shown in Table 2.

**Table 2.** Effect of covariance matrix update method on continuous point sources.

| Index | Unit | Standard | Fixed Step | Memory Matrix | Mix |
|---|---|---|---|---|---|
| Proposed sample total acceptance rate tar | % | 59.91 | 64.55 | 61.52 | 60.39 |
| Iteration time-consuming eti | s | 368.66 | 80.87 | 79.67 | 76.59 |
| Execute time-consuming ete | s | 743.12 | 421.32 | 432.72 | 449.31 |

It can be seen from Table 2 that in the continuous point source tracking simulation, the iteration time of the fixed step size, memory matrix and chaos improvement method is reduced by 78.06% compared with the traditional method. The memory matrix is reduced by 78.39%. The hybrid improved method is reduced by 79.22%. In terms of implementation time, the completion time is reduced by 43.90%, 43.23% and 39.84%, respectively. At the recommended overall acceptance level, the overall acceptance level is higher, which is at

the better level. The influence of the covariance matrix update method on the intermittent point source when Nc = 1 is shown in Table 3.

**Table 3.** Influence of covariance matrix update method on intermittent point source when Nc = 1.

| Index | Unit | Standard | Fixed Step | Memory Matrix | Mix |
|---|---|---|---|---|---|
| Proposed sample total acceptance rate tar | % | 52.41 | 52.24 | 60.85 | 60.33 |
| Iteration time-consuming eti | s | 808.58 | 544.84 | 531.28 | 531.23 |
| Execute time-consuming ete | s | 1213.97 | 923.80 | 867.50 | 885.08 |

It can be seen from Table 3 that in the retrospective simulation of discontinuous point source (Nc = 1), the iterative consumption time of fixed step size, memory matrix and improved chaos algorithm is reduced by 32.62%, 34.29% and 34.30% compared with the conventional method. The results show that the implementation time of this method is shortened by 23.80%, 25.75%, and 24.95%, respectively. The results show that, except that the overall acceptance rate of sampling decreased with the new method, the other two methods have improved, which are all within the better interval. The influence of the covariance matrix update method on the intermittent point source when Nc = 3 is shown in Table 4.

**Table 4.** Influence of covariance matrix update method on intermittent point source when Nc = 3.

| Index | Unit | Standard | Fixed Step | Memory Matrix | Mix |
|---|---|---|---|---|---|
| Proposed sample total acceptance rate tar | % | 48.38 | 43.40 | 76.36 | 74.87 |
| Iteration time-consuming eti | s | 2847.40 | 2379.19 | 2498.04 | 2923.10 |
| Execute time-consuming ete | s | 3343.03 | 3128.38 | 2978.38 | 3432.43 |

From Table 4, in the retrospective simulation of discontinuous point source (Nc = 3), the repeated consumption time of fixed step size and memory matrix update method are 16.47% and 13.52% and 2.87%, respectively. The results show that the time of adopting the hybrid update method is reduced by 6.76% and 12.09%, respectively. Except for the new fixed-step method, the overall acceptance rate of the sampling is reduced. The other two methods are in the higher interval.

## 4. Results and Discussion

The application of artificial intelligence (AI) in this study improved the qualitative boron isotopic and Na/B molar ratios in the water bodies of the Fen River Basin compared with retrospective methods. In recent years, the river environment problems in the Fenhe area have become increasingly serious. In this paper, artificial intelligence technology is used to study the source tracing under boron isotope. Among them, more than half of the water quality is dominated by secondary water quality, which has become a "bottleneck" affecting the economic development of the region. Further refinement of the ecological function zoning and industrial structure layout of the Fenhe River Basin, promoting the ecological restoration of mining areas, and realizing the green and water-saving transformation of coal mining are all recommended strategies. In order to solve the problem of river water pollution in the Fenhe River Basin, this paper has collected a large amount of river water, rainwater and possible pollution sources in the wet season and dry season of the Fenhe River Basin. Combined with boron isotopes, the geochemical characteristics of pollutants in the river water of the Fen River Basin have been explored. Disintegrated B in waterways mostly comes from silicates, evaporites, fake data sources, environmental data sources, and carbonates. In the dry season, the contribution of precipitation to B was 7.8%, 16.4%, 37.3% and 38.8%, respectively. During the wet season, the contribution of B to the air

was 6.7%, 15.0%, 27.1% and 49.2%, respectively. It has laid the foundation for the ecological environment protection and pollution prevention and control work in the Fenhe area.

**Author Contributions:** Conceptualization, G.H., H.Y. and Z.Y.; methodology, G.H.; software, Z.Y.; validation, G.H.; formal analysis, H.Y.; investigation, Z.Y.; data curation, Z.Y.; writing—original draft preparation, G.H.; writing—review and editing, H.Y.; supervision, H.Y.; project administration, H.Y. All authors have read and agreed to the published version of the manuscript.

**Funding:** This work was supported by the National Natural Science Foundation of China: application of boron isotopes for tracing source of contamination in Shayinghe river (No. 42173062).

**Data Availability Statement:** The experimental data used to support the findings of this study are available from the corresponding author upon request.

**Conflicts of Interest:** The authors declared that they have no conflict of interest regarding this work.

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
