# Peer review of "Application of AI Identification Method and Technology to Boron Isotope Geochemical Process and Provenance Tracing of Water Pollution in River Basins"

_sustainability, doi:10.3390/su15075942_

Round 1

Reviewer 1 Report

Reviewer Comment: Article No. sustainability-2180866

Application of AI Identification Method and Technology to Boron Isotope Geochemical Process and Provenance Tracing of 2 Water Pollution in River Basins 3

 Abstract

Abstract must consist of Introduction, Problem Statement, Methodology, Result & Discussion and Conclusion/ Final Findings.

The statement after “The quantitative analysis of water quality is even more difficult” seems incomplete. Why the quantitative is difficult shall be supported. The problem statement stated “Since the 20th century, 10 human influences on river courses has become increasingly serious” was not clearly explained. Are the 10 influences related to the anthropogenic activities?

The author is required to state the methodology used in this study. And how the collection of the pollution been conducted, and monitoring period/ study duration (covering how many years) are also needed to be mentioned herein.

For determining the physico-chemical properties and pollutants in the water body of the Fenhe River, what type of the lab analysis conducted. And the various parameters of pollutants need to be clearly highlighted as well.  Then, through the statistical analysis the most significant parameters or variables would be determined.

Regression-based predictive relationships are not sufficed for this analysis. The analysis should encompass of the trend analysis enable the determination of substantial physico-properties of the pollutants.

Apparently, there is no conclusion or final findings in the abstract.

Introduction

Most of the citations are acceptable within recent five years. Upon reviewing I realize that some of the reference seems not related to this manuscript.

Line 41 – to highlight the source of pollution either point source or non-point source. Is the study been spatially conducted for water quality pollution?

Line 27-50 the statements are not supported with the citation. Please cite the relevant papers related to the manuscript. The sentence “Therefore, the quality of rivers has been deeply explored and quantitatively analyzed.”, this statement is it refers to the pollutant parameters? Please clearly explain with supported by the previous research.

The sentence “it is difficult for traditional methods to effectively trace its source.” Please state type of methods. Is kind of vague.

Line 82-83. The statement “He also carried out qualitative and quantitative analysis of DWT.” Please state the full name of DWT.

Since this study involves with artificial intelligent (AI), I perceived there is no argument or highlight on it. The second last paragraph should discuss on the significant role of AI in this study. Unfortunate, the author did not stress at all. Apparently, the statistical analysis of correlation analysis, principal component analysis, multi-factor regression analysis as you claimed used in this study for predictive trend of physico-chemical properties of the pollutants has never been stated elsewhere in this paragraph too. Did the author use the multivariate or chemometric techniques to carry out the qualitative analysis? The quantitative analysis has left out in the introduction.

The last paragraph of the introduction the author requires to include the findings of the study. 

Materials and Methods

The materials and methods are not stated in the manuscript.

When the sampling period was carried out. The study should be conducted at least covering a period of one year based on rainfall intensity and duration to study the trend or pattern to of physico-chemical characteristics of traceability elements, the influencing factors of pollutants,etc.  Is not mentioned in the manuscript!!!!

How been the quantitative analysis carried out? There is not availability of the analytical laboratory analysis in this manuscript!!!!

2. Geochemical Process of Boron Isotope in Water Pollution and Provenance Tracing 104

Method

Please state the integration of AI and boron isotopic in traceability elements of pollution in this study. Not being highlighted at all. Straight to the point nor browsing around the bushes.

3.1 Overall Design of Water Pollution Traceability Management System

The structural framework of water pollution traceability management system did not include the AI system. I do not comprehend with this as no explanation on the nexus AI with baron isotopics.

3.2. Quantitative Analysis of the Sources of Pollutants in the Fenhe River Basin

Where is the quantitative analysis? How were the samples being brought to the lab and what temperature should store the samples? And were the samples analyzed straight away or stored in the freezer at certain period and temperature?

In research, when we conduct the laboratory analysis, we must analyze all the parameters found in the samples. Each sample must be replicated for analysis for precision and accuracy.

The author to highlight on the origin of the Fenhe River, the boron isotope content in the wet season and dry season that influence of physico-chemical characteristics, that its boron content is closely related to rainfall (Line 311). The research on these need to be elaborated as well. Difference in intensity and duration impacted the types and number of pollutants in the river. Upstream, near the end members of the gasification rock, the boron content is very low. Please clarify. Also, the water quality of the central reaches is similar to the end elements of agricultural pollution, indicating that the agricultural environmental pollution in this region has a certain contribution to the water quality of the watershed. Predictive trend analysis must be carried out for all three influencing factors, for wet and dry seasons covering the areas of boron content upstream and agricultural area that related boron in the soil of the Fenhe River Basin.

Results and Discussion

Where is the results and discussion section?

The results and discussion are not comprehensively discussed.

The statement “Due to the high boron content in the soil of the Fenhe River Basin, the boron isotope level is high after entering the Yellow River.” Please elaborate the reasons.

The first statement of results and discussion should state that the application of Artificial Intelligence (AI) in this study have improved the qualitative boron isotopes and Na/B molar ratio in the water body of the Fenhe River Basin compared to the retrospective approach.

The author did not discuss on the analysis of ANN in this section. Please elaborate.

Analysis of data using the principal component analysis (PCA) can be executed to obtain the most significant variables found in the qualitative boron isotopes and Na/B. Also, should be highlighted the causal factors as well attributable to the scenario. What is the chemical process taken place in the rainfall as resulted from the urbanization, such as smokes, surface runoff due to site clearance for residential areas, etc; point sources, instantaneous point source and non-p0int source. Therefore, the study will be meaningful and promising to support the findings. Simple linear regression between the parameters is unable to indicate a significant overall correlation of the influencing factors of various of pollutants in the storm water.

This statement “During the wet season, the contribution of salinity to 389 B is 23.7±15.1% (1 SD). The upstream, midstream, and downstream contribute 16%, 31%, 390 and 30%, respectively. The study found that in the rich water period, the contribution of 391 gypsum dissolution to B is 3.1-4.3%. Shahe accounts for 0.1%. The interactive river con-392 tributes the most (23.4%). In the dry season, the contribution to B reaches 4.3±5.9% (1 SD). 393 In arid regions, the contribution to the dissolution of gypsum is basically the same as that 394 in the wet season. Shahe accounts for 0.1%. Sediment contributes the largest, accounting 395 for 31.7%. The general view is that sediment and sediment dissolution can reflect the effect 396 of total evaporation on B.”, requires profound interpretation!!!!! What I notice none of statement in this section discussing the influencing factors impacted the instantaneous point source, etc.

The analysis also needs to carry out the spatial and temporal the trend analysis of the boron isotope and highlight the correlation of boron isotopes and Na/B in rainwater and possible pollution sources in the wet season and dry season of the Fenhe River Basin.

Conclusion

The conclusion is acceptable.

References

Please follow the APA standard.

Reviewer 2 Report

The manuscript needs to be improved, as shown below: 

1. The abstract may be improved by providing more specific details on the results and findings of the study. For example, it could give more information on the extent of pollution and the specific sources identified. It could also include a summary of the conclusion and the implications of the research, specifically how it can help protect water resources and control pollution in the Fenhe River Basin.

2. The introduction section does not provide a clear problem statement and the overall flow of the text is disjointed, making it hard to understand. The information about the various studies mentioned is presented disconnected, and there is a lack of clear purpose and coherence. To make the paper more accessible and understandable, the authors may consider revising the introduction to provide a clear problem statement and a more organized overview of the relevant studies and their findings.

3. Please add a comparative table of relevant studies against this work in the introduction section to help clarify the study contributions by highlighting similarities, differences, and strengths and weaknesses of the current work compared to previous research.

4. Section 2.1 lacks clarity and accuracy. Please revise accordingly. Revise the section on the connection between intelligence and computers to ensure clarity and conciseness. Consider incorporating the main points about the two levels of AI, its connection to multiple fields, and the ongoing advancements and expansion of AI research and application. Ensure the paragraph is connected to the previous one for better flow and coherence.

5. Line 162: Where is the variable a? The equation earlier does not have such a variable. Please be more specific and revise accordingly. Section 2.2 needs to be revised for clarity, as there seems to be no connection between paragraphs and a lack of description.

6. Line 176: There is a mismatch in the referred equation. Please also define all symbols used in all equations.

7. The current conclusion lacks an explicit description of the application of AI in the study. While it provides a general overview of the results, it fails to specify the role of AI in generating these results. It is recommended that the authors revise the conclusion to incorporate a technical description of the AI methodology employed in the study, including the specific algorithms and techniques used and how they contributed to the results. This will provide greater detail and a more nuanced understanding of the role of AI in water pollution analysis and its impact on the geochemical process.

Round 2

Reviewer 1 Report

The author must revise this paper accordingly, based on the comments given.  

Reviewer 2 Report

non

Round 3

Reviewer 1 Report

The author have revised based on comments.